# A Novel Supervised Filter Feature Selection Method Based on Gaussian Probability Density for Fault Diagnosis of Permanent Magnet DC Motors

**DOI:** 10.3390/s22197121

**Published:** 2022-09-20

**Authors:** Weihao Wang, Lixin Lu, Wang Wei

**Affiliations:** School of Mechatronic Engineering and Automation, Shanghai University, 99 Shangda Road, BaoShan District, Shanghai 200444, China

**Keywords:** feature selection, fault diagnosis, dimension reduction, machine learning

## Abstract

For permanent magnet DC motors (PMDCMs), the amplitude of the current signals gradually decreases after the motor starts. In this work, the time domain features and time-frequency-domain features extracted from several successive segments of current signals make up a feature vector, which is adopted for fault diagnosis of PMDCMs. Many redundant features will lead to a decrease in diagnosis efficiency and increase the computation cost, so it is necessary to eliminate redundant features and features that have negative effects. This paper presents a novel supervised filter feature selection method for reducing data dimension by employing the Gaussian probability density function (GPDF) and named Gaussian vote feature selection (GVFS). To evaluate the effectiveness of the proposed GVFS, we compared it with the other five filter feature selection methods by utilizing the PMDCM’s data. Additionally, Gaussian naive Bayes (GNB), k-nearest neighbor algorithm (k-NN), and support vector machine (SVM) are utilized for the construction of fault diagnosis models. Experimental results show that the proposed GVFS has a better diagnostic effect than the other five feature selection methods, and the average accuracy of fault diagnosis improves from 97.89% to 99.44%. This paper lays the foundation of fault diagnosis for PMDCMs and provides a novel filter feature selection method.

## 1. Introduction

### 1.1. Research Motion

Permanent magnet DC motors (PMDCMs) are widely used in automobiles, household electrical appliances, industrial production, and so on. PMDCM faults may cause noise, vibration, and mechanical damage, which will result in economic losses and poor user experience. Thus, it is necessary to carry out a comprehensive fault diagnosis for PMDCMs before they are out of the factory. However, artificial diagnosis easily leads to misjudgment and is time-consuming. Note, that with the rapid development of computer, communication, and storage technologies, artificial intelligence (AI) is an effective way to solve this problem. However, the increase in the amount of data will cause a tremendous burden on communication, which causes communication data loss and communication delay. Gu et al. [1] found that the communication delay is in gamma distribution [2], and researched distributed tracking control of networked control systems, load frequency control of power systems, and image encryption [3], respectively, and used servo motor experiments to verify the effectiveness of the proposed method. To release the communication burden, a novel weighted integral event-triggered scheme (IETS) is proposed by Yan et al. [4], and an application to image encryption is utilized to demonstrate the advantage of the proposed IETS. Moreover, this paper proposes a novel supervised filter feature selection method to release the communication burden.

In this paper, Gaussian naive Bayes (GNB), k-nearest neighbor algorithm (k-NN), and support vector machine (SVM) are utilized for online fault diagnosis of PMDCM to improve the accuracy of fault diagnosis. Considering the efficiency of online fault diagnosis, this paper proposes a novel supervised filter feature selection method based on Gaussian probability density function (GPDF) for fault diagnosis of PMDCMs to improve the fault diagnosis efficiency while ensuring fault diagnosis accuracy.

### 1.2. Literature Review of Feature Selection

Feature selection has been widely used in text classification [5,6], speech emotion recognition [7,8], etc. K et al. [5] proposed a hybrid feature selection method for text classification based on a binary poor and rich optimization algorithm (HBPRO). Cekik et al. [6] proposed a feature selection method called a proportional rough feature selector (PRFS) for the high dimensionality problem of short text. Yildirim et al. [7] modified the initial population generation stage of metaheuristic search algorithms to reduce the number of features for emotion classification from speech. For the hierarchical classification problem, Liu et al. [9] proposed a hierarchical feature selection method with a capped l2-norm (HFSCN), which can learn how to discriminate relatively robust feature subsets. Wan et al. [10] considered the interaction properties of rough neighborhood sets. They proposed a feature selection method to reduce the dimensions of hybrid data with uncertainty and noise. Abasabadi et al. [11] proposed a three-step ensemble feature selection method, named automatic thresholding feature selection method (ATFS), a filter method based on interference between class labels. It focuses on the information between not adjacent data pairs. Min et al. [12] proposed the multi-step Markov probability relationship for feature selection (MMFS), an unsupervised feature selection method that utilizes multi-step transition probability to characterize the data structure. Zhou et al. [13] focused on feature interaction between the streaming subsets. They proposed an online group streaming feature selection method (OGSFS-FI), which selects features that can communicate with each other. Wang et al. [14] jointed learns soft-labels and feature selection matrix to obtain an effective unsupervised soft-label feature selection (USFS) method. The USFS model can eliminate the features with negative effects and overcome the intrinsic semantic limitations of unsupervised feature selection methods. Salem et al. [15] proposed a fuzzy joint mutual information (FJMI) feature selection method that automatically selects the best feature subset automatically without a user-defined threshold. Gao et al. [16] proposed a relevance assignation feature selection (RAFS) method based on mutual information (MI), which can select the significant features and deny the undesirable ones. Lall et al. [17] proposed copula-based feature selection (CBFS) to discover the dependence structure between features utilizing a multivariate copula. To improve efficiency and classification accuracy, Zhou et al. [18] proposed a feature selection method of decision trees based on feature weight, called FWDT. Dong et al. [19] presented a multi-label feature selection method (MMFS) based on many-objective optimization for multi-label classification and feature selection problems. Wei et al. [20] proposed a method to measure the relationships of features accurately named dynamic feature importance (DFI). Based on DFI, they proposed a feature selection method named dynamic feature importance feature selection (DFIFS). To obtain higher accuracy with a smaller number of features, the Modified-DFIFS method is developed by combining DFIFS with the classical filters.

### 1.3. Literature Review of Fault Diagnosis

For rotating machines and equipment, vibration signals, electrical signals and acoustic signals are widely used for fault diagnosis and condition monitoring of bearings, gears and motors in combination with machine learning methods [21]. In this paper, the current signal is taken as the original signal, the GNB, k-NN and SVM are utilized as classifiers, and six feature selection methods are combined to diagnose the fault of PMDCMs.

Altinors et al. [22] diagnosed the fault in the brushless DC (BLDC) motors used in Unmanned Aerial Vehicles (UAV), by utilizing the sound data received from the motors, the k Nearest Neighbor (k-NN), Decision tree (DT), Support Vector Machines (SVM) methods are used for the construction of the fault diagnosis models. Jiang et al. [23] proposed a strong robustness diagnosis strategy based on d-q-axis current signal to diagnose the Open-Circuit (OC) fault in the novel fault-tolerant electric drive system. The result shows that the proposed OC fault diagnosis strategy can overcome the OC fault diagnostic false alarms issue when the load changes suddenly or under light-load conditions. Allal et al. [24] proposed a residual harmonic analysis method by normalizing the motor current to predict faults and avoid unexpected failures of the induction motor. Li et al. [25] proposed a novel fault diagnosis method by comparing the trajectory distribution characteristics of different open-circuit faults for the single-phase (two-phase) open-circuit faults of the five-phase permanent magnet synchronous motor (FP-PMSM). They built a test platform with a DSP core to verify the proposed fault diagnosis method. Zhang et al. [26] proposed a time-shifting based hypergraph neural network (TS-HGNN) to diagnose the fault of electromechanical coupled systems by considering the association relationship between multiple nodes of graph-based networks. Zhang et al. [27] proposed a novel conception of inter-turn short circuit fault and utilized the Hidden Markov Model to diagnose the inter-turn short circuit fault of DC brushed motors. The validity of the proposed method for inter-turn short circuits of DC brushed motors is verified by simulating in MATLAB/Simulink. Chen et al. [28] believed that the analysis of motor current characteristics is a very promising technology in electromechanical systems and proposed a fault diagnosis method for planetary gearboxes under variable speed conditions by using stator current signal.

### 1.4. Contributions and Novelties

As for the previous research on fault diagnosis, lots of fault diagnosis methods for motors have been proposed based on vibration signals, electrical signals and acoustic signals, by utilizing the machine learning method. The online fault diagnosis of PMDCM not only needs to ensure diagnosis accuracy but also needs to ensure diagnosis efficiency. Therefore, this paper proposes a novel feature selection method, which is combined with machine learning to ensure diagnosis efficiency by reducing the dimension of features. As for dimension reduction, lots of methods have been proposed so far. However, some are applied to domain-specific [5,6,7,8], some are aimed at solving high-dimension problems [29], some are to solve the problem of unbalanced samples, some are focused on the relationship between the features [13,30], and some are adopted heuristic algorithms. They are limited to a particular field or condition. Thus, we proposed a novel supervised feature selection method whose features only need to be numerical, and suitable for the fault diagnosis of PMDCMs. The main contributions of the paper are summarized as follows.

In the study, current signals are utilized for the fault diagnosis of PMDCMs by utilizing the GNB, k-NN, and SVM classifiers, successively. To improve the fault diagnosis efficiency of PMDCMs, the paper proposed a supervised filter feature selection method based on GPDF.

The proposed fault diagnosis method of PMDCM in this paper not only improves the diagnostic accuracy and efficiency, but also weakens the dependence on manual experience, and improves the reliability of diagnosis results, which has potential commercial promotion and application value.

### 1.5. Study Outline

The rest of the paper is organized as follows: Section 2 describes the background of feature selection and Gaussian probability density and introduces the Gaussian probability density function (GPDF). Section 3 introduces the proposed feature selection methods based on the GPDF. Section 4 introduces the experiment framework in this paper, the test rig, the feature extract, and the other five filter feature selection methods. Section 5 shows the fault diagnosis results of PMDCMs and makes an analysis. Finally, Section 6 presents conclusions and ideas for future work.

## 2. Background

### 2.1. Feature Selection

Feature selection aims to select the most applicable features while ignoring the irrelevant and redundant ones, and feature selection methods are generally divided into three types: filter, wrapper and embedded [6,10,11,20,29,30].

The most widely used filter method is easy to implement and can obtain good performance. The filter methods usually are defined as a scoring mechanism for features, then, the scores of features are calculated, and features with below threshold scores are filtered out. The embedding feature selection method can be regarded as an evolutionary filter feature selection method, which embeds the classifier into the process of feature selection. The evaluation metrics are utilized as the weight of the feature, and then filter out features with weights below the threshold. The wrapper feature selection method selects a subset of features by using a classifier and evaluates the feature subset with a metric (e.g., accuracy, precision, and recall) in each cycle. After several rounds of selection, the best subset of features can be obtained. The disadvantage of the wrapper feature selection method is computing-intensive and time-consuming.

Feature selection methods can be classified as supervised [31], semi-supervised [32] and unsupervised [12,14], on the basis of whether labels are used or not. Supervised feature selection methods make full use of labels in training and testing and pay more attention to the relationship between features and labels. For semi-supervised feature selection methods, most of the data are unlabeled. The data of unsupervised feature selection methods are not labeled, and unsupervised feature selection methods only consider the characteristics of features and the connections between features.

The proposed GPDF based feature selection method is a supervised filter feature selection method. For more details, see Section 3.

### 2.2. Gaussian Probability Density

If a random variable X follows a Gaussian distribution with mathematical expectation μ and variance $\sigma^{2}$, denoted by $X\sim N\left(\mu ,\sigma^{2}\right)$, the Gaussian probability density function (GPDF) is as in Equation (1).
(1)$$f\left(x\right)=\frac{1}{\sqrt{2\pi }\sigma }\exp\left(-\frac{{\left(x-\mu \right)}^{2}}{2\sigma^{2}}\right)$$

As can be seen from Figure 1, x=μ is the symmetry axis of the GPDF, which determines the central position of the Gaussian PDF. The variance $\sigma^{2}$ describes the centralization degree of the GPDF. The larger the $\sigma^{2}$, the more dispersed the data are distributed. Figure 1 shows four sets of the GPDF function: $\mu =0,\;\sigma^{2}=0.5;\;\mu =0,\;\sigma^{2}=1;\;\mu =0,\;\sigma^{2}=2;\;\mu =2,\;\sigma^{2}=0.5$. It can be seen that the GPDF has the same symmetry axis for a given same value of μ. When the GPDF has the same $\sigma^{2}$, they have the same distribution curve, and the curve of the GPDF gradually becomes smooth with the increase of $\sigma^{2}$.

The Gaussian probability function (Equation (2)) in the interval [a,b] is derived from Equation (1).
(2)$$P\left(a<x<b\right)=\int_{a}^{b}f\left(x\right)dx=\frac{1}{\sqrt{2\pi }\sigma }\int_{a}^{b}exp-\left(\frac{{\left(x-\mu \right)}^{2}}{2\sigma^{2}}\right)dx$$
where P(·) represents the probability value of x in the interval [a,b], a represents the lower limit of the GPDF, and b represents the upper limit of the GPDF. Integrate the GPDF in the interval [a,b], to represent the probability value of the GPDF in the interval [a,b] (i.e., Calculate the area under the GPDF curve in the interval [a,b]). Therefore, the probability value of x can be approximately regarded as the Integrate from x to x+Δx, and can be calculated by Equation (3).
(3)$$P_{x}=\int_{x}^{x+\Delta x}f\left(x\right)dx=\frac{1}{\sqrt{2\pi }\sigma }\int_{x}^{x+\Delta x}exp\left(-\frac{{\left(x-\mu \right)}^{2}}{2\sigma^{2}}\right)dx\approx \Delta x\times f\left(x\right)$$
where $P_{x}$ represents the probability value of x.

Suppose that, there are four GPDFs (e.g.,$\;f_{1}\left(x\right),\;f_{2}\left(x\right),f_{3}\left(x\right),\;f_{4}\left(x\right)$), and the probability values of on $\;f_{1}\left(x\right),\;f_{2}\left(x\right),f_{3}\left(x\right),\;f_{4}\left(x\right)$ are expressed as $P_{1},P_{2},P_{3},\;P_{4}$, respectively, the ratio of $f_{1}\left(x\right),\;f_{2}\left(x\right),f_{3}\left(x\right),\;f_{4}\left(x\right)$ can be expressed by Equation (4).
(4)$$P_{1}:P_{2}:P_{3}:P_{4}=f_{1}\left(x\right)\;:\;f_{2}\left(x\right)\;:\;f_{3}\left(x\right)\;:f_{4}\left(x\right)$$

As shown in Figure 1, when x=1, $P_{1}:P_{2}:P_{3}:P_{4}=f_{1}\left(1\right):f_{2}\left(1\right):f_{3}\left(1\right):f_{4}\left(1\right)=0.10789:0.24197:0.17603:1.2\times 10^{-8}$.

## 3. Proposed Feature Selection Method

### 3.1. Feature Structure Introduction

In this paper, a supervised feature selection method based on the GPDF is proposed. In this section, the derivation process of the proposed feature selection method is described in detail. Given a feature space $T=\left\{\left(x_{1},y_{1}\right),\left(x_{2},y_{2}\right),\cdots ,\left(x_{N},y_{N}\right)\right\}$, where $x_{i}\in R^{d}$,$\;y_{i}\in \left\{1,2,\cdots ,l\right\}$, i=1,2,⋯,N, $x_{i}$ is the i-th sample, with label $y_{i}$. Sample $x_{i}$ is a real vector of d-dimension features, denoted by $x_{i}=\left(x_{i}^{\left(1\right)},x_{i}^{\left(2\right)},\cdots x_{i}^{\left(d\right)}\right)$, the feature structure of the dataset is shown in Table 1.

### 3.2. Gaussian Vote Feature Selection

The GVFS proposed in this paper is suitable for classification problems in which the number of categories is greater than or equal to two. The specific steps to implement the feature selection method based on GPDF are as follows:

Step 1: Normalizing each feature to [0, 1];

Step 2: Dividing the samples set into the training dataset and test dataset;

Step 3: Dividing each feature of the training dataset into multiple subsets according to the labels of samples;

Step 4: Calculate the Gaussian parameters ($\mu \;\mathrm{and}\;\sigma^{2}$) of each subset separately;

Step 5: Calculate the Gaussian probability density function (GPDF) value for each feature of each category;

Step 6: Predict the category of each feature ($f_{1},\;f_{2},\cdots ,f_{d}$) so that each sample corresponds to d (feature dimension) prediction categories $y_{fd}$;

Step 7: Calculate the prediction accuracy of each feature $Acc_{fd}$ by comparing the prediction category $y_{fd}$ with the actual category $y_{N}$;

Step 8: Sort the features in descending order, according to the prediction accuracy of each feature $Acc_{fd}$;

Step 9: Select the first S features to make up the final feature subset.

Figure 2 shows the process of feature category prediction based on the GPDF. All samples must be normalized, and the training samples are utilized to calculate the GPDF parameters ($\mu \;\mathrm{and}\;\sigma^{2}$) for each feature of each category. For the example in Figure 2, the samples are divided into four categories ($y_{1},\;y_{2},\;y_{3},\;y_{4}$), and the red lines represent the feature values of the sample that is to be tested. The GPDF parameters ($\mu \;\mathrm{and}\;\sigma^{2}$) for each category are calculated for each feature ($f_{1},\;f_{2},\cdots ,f_{d}$) based on the feature values of the tested sample. For each feature, the category with the maximum GPDF $\left(max\left\{f_{y1}\left(x\right),\;f_{y2}\left(x\right),\;f_{y3}\left(x\right),\;\;f_{y4}\left(x\right)\right\}\right)$ is adopted as the prediction category ($y_{f1},\;y_{f2},\;\cdots ,\;y_{fd}$), $y_{fd}\in \;\left(y_{1},\;y_{2},\;y_{3},\;y_{4}\right)$, and the accuracy $Acc_{fd}$ corresponding to each feature is calculated when all the test samples have been classified, then the features are sorted according to the accuracy $Acc_{fd}$.

The pseudo code of the proposed GVFS method is shown in Algorithm 1. The labeled data set is taken as input, and the feature sorted dataset as the output. The pseudo code of the GVFS consists of two parts: in the first part, the GPDF parameters are calculated by training samples. In the second part, predict the category of each feature of the test sample by utilizing the constructed GVFS model, and the features are sorted according to the prediction accuracy of the features. The process of the proposed GVFS method can be seen in Figure 3.
**Algorithm 1** GVFS methodInput: dataset X with label Y
Output: feature sorted dataset R with label Y
1: X = normalize (X);2: Dividing the samples set into the training dataset and test dataset;3: Calculate the $\mu \;\mathrm{and}\;\sigma^{2}$ for each feature on each category of the training dataset [33].4: for i = 1:1:sample_number5: Calculate the Gaussian probability for each feature on each category [33].6: The category with the largest GPDF is taken as the category for each feature.7: end8: Calculate the prediction accuracy for each feature.9: The features are sorted (R) according to the prediction accuracy of each feature.

## 4. Experiment Setting

This study aims at constructing an effective and feasible fault diagnosis model for permanent magnet DC motors (PMDCMs) based on Gaussian naive Bayes (GNB) [33], k-nearest neighbor algorithm (k-NN) [34], and the support vector machine (SVM) [35] by utilizing the filter feature selection methods.

### 4.1. Experiment Framework

The experiments were carried out on a PC with Intel Core i5-8250U, 1.8-GHz CPU, 24 GB RAM, 1 TB hard disk, and Windows 11 operating system. Python 3.8 is utilized to process the signal and construction models.

The framework of the experiment is shown in Figure 4. Firstly, the experimental platform is introduced, and then 1200 groups of PMDCM’s current signal are collected by LabVIEW 2018 software. Then, the time-domain features and time-frequency-domain features are extracted from the successive multi-segment PMDCM’s current signal. The extracted features are normalized by utilizing the Min-Max normalization. A GNB classifier is used to optimize the number of iterations and the proportion of test samples for normalized features. After determining the number of iterations and the proportion of test samples, the proposed GVFS method and five other feature selection methods (in Section 4.6) are used to sort the extracted features, and k-NN classifiers with different k values are used to evaluate the different numbers of features, respectively, so as to obtain the number of features with the optimal fault diagnosis accuracy and the number of features with the sub-optimal fault diagnosis accuracy. Optimization of the two hyperparameters (C and $\sigma^{2}$) of the SVM classifier by utilizing the 70% samples with selected features is conducted; at the same time, 10-fold cross-validation is utilized to stabilize the performance. Then, training of the SVM fault diagnosis model is conducted with the optimized two hyperparameters and 70% samples, the remaining 30% samples are used to test the fault diagnosis accuracy of PMDCMs.

### 4.2. Test Rig Introduction

As can be seen from Figure 5, the test rig consists of a computer, a control cabinet, and tested PMDCMs. The computer is utilized for data collection, data processing, model construction, result display and so on. The control cabinet consists of the following: 1. The 24 V power supply, which is utilized to provide the 24 V power source for the relays and data acquisition card; 2. The 12 V power supply, which is utilized to provide the 12 V power source for PMDCMs; 3. The front-end relay, which is utilized to control the power on and power off for the data acquisition card; 4. The timer relay, which is utilized to relay the power on for PMDCMs; 5. The four-channel relay, which is utilized to control the power on and power off for PMDCMs; 6. The sampling resistance is utilized to convert the current signal to the voltage signal. 7. The data acquisition card is utilized to collect the voltage signal from the sampling resistance and save the voltage signal to the computer. For more detailed information on the control cabinet parts, please find them in Table 2.

### 4.3. Data Collection

In this work, the current signal of PMDCMs is collected by transforming the current signal into the voltage signal. The sampling frequency is set to 10k Hz, and the duration of the current signal is set to 10 s. The collected current signal data of PMDCMs are saved in the format of LVM by utilizing the software LabVIEW 2018.

This research subject comes from the production demand, and the fault diagnosis problem of PMDCM has been put forward by the manufacturing enterprise. In order to detect and repair the fault of PMDCMs in time, it is necessary to implement the final inspection of quality before the PMDCMs leave the factory. The utilized PMDCMs in this study are from motor manufacturers, and the faults of PMDCMs are natural. According to the experience of motor manufacturers, the PMDCM’s faults are divided into six categories, as can be seen in Table 3. The faults are natural failures in the production process rather than man-made failures. The motor will vibrate due to shaft unbalance. The noise caused by bearing friction and the bearing slipping will produce obvious intermittent noise and slight vibration.

As can be seen from Table 3, the PMDCMs are divided into six categories (healthy, slight noise, loud noise, harsh noise, shaft unbalance, and bearing slipping), and 200 samples are collected for each category of PMDCM, a total of 200×6=1200 samples are obtained.

### 4.4. Feature Extraction

Features need to be extracted from the current signals with the aim to reflect the PMDCM’s health state. Feature extraction involves deriving time-domain, frequency-domain, and time-frequency-domain features from the raw signals. In this work, wavelet packet decomposition (WPD) [36] is employed for the extraction of the time-frequency features. The raw signals are decomposed up to three levels. The detailed coefficients and approximate coefficients of levels 1 to 3 are obtained and from which the wavelet energy is computed. Moreover, nine time-domain features are extracted from the current signals for PMDCMs fault diagnosis. A total of 17 features, 9 time-domain and 8 time-frequency-domain are extracted from the signals. The time domain features are presented in Table 4.

As can be seen from Figure 6, the current signals of each type of PMDCM are different in both starting and stable stages [37]. To diagnose the health condition of PMDCMs effectively, the current signals are divided into multi-segment and extracted the 17 features for several successive segments, respectively, and the extracted features are adopted for fault diagnosis of PMDCMs. We divided the current signals into 16 segments, the first 12 segments start from 0.5 s with a duration of 0.025 s, and the last four segments with a duration of 0.5 s, for each segment to extract 17 features, a total of 16×17=272 features are extracted for each sample. (s represents the second.)

### 4.5. Feature Preprocessing

Features are transformed into the same order of magnitude, after normalizing. so that different features have the same weight, which can be comprehensively evaluated and analyzed. At the same time, normalization of the data can avoid numerical problems caused by large values, so the features need to be normalized before model construction. The commonly used normalization methods are Min-Max normalization and Zero-Score normalization, the equations are as shown in Equations (5) and (6).
(5)$$\tilde{x_{i}}=\frac{x_{i}-min\left(x\right)}{max\left(x\right)-min\left(x\right)}$$
(6)$$\tilde{x_{i}}=\frac{x_{i}-\mu }{\sigma }$$
where min (·) and max (·) denote the operation that returns the minimum and maximum values of x, $\tilde{x_{i}}$ represents the new value of $x_{i}$. μ represents the mean value of $x_{i}$, σ represents the standard deviation of $x_{i}$. The Min-Max normalization method transfers each feature to the interval of [0, 1], and only scales the features equally. However, the Zero-Score normalization method maps the original features to a standard Gaussian distribution, which changes the distribution of features. Therefore, the Min-Max normalization method is utilized to normalize the features.

### 4.6. Feature Selection Methods

In this work, the proposed GVFS method is compared with the other five feature selection methods (e.g., Fisher score feature selection (FSFS) [38], mutual information feature selection (MIFS) [10,15,39], variance threshold feature selection (VTFS) [40], F-test feature selection (F-testFS) [41], chi-square feature selection (chi2FS) [42]) by utilizing the PMDCM’s data.

(1)FSFS

Fisher score feature selection (FSFS) [38] is one of the effective filter methods for feature selection. The main idea of FSFS is that find the features with the distances between different categories being as large as possible, while the distances of the same category are as small as possible. Specifically, let $\mu_{l}^{j}$ and $\sigma_{l}^{j}$ be the mean and standard deviation of l-th category, corresponding to the j-th feature. Let $\mu^{j}$ and $\sigma^{j}$ denote the mean and standard deviation of the j-th feature. c represents the number of categories. $n_{l}$ represents the number of samples with the label l. Then the Fisher score of the j-th feature is computed according to Equation (7).
(7)$$F\left(x^{j}\right)=\frac{\sum_{l=1}^{c}n_{l}{\left(\mu_{l}^{j}-\mu^{j}\right)}^{2}}{\sum_{l=1}^{c}n_{l}{\left(\sigma_{l}^{j}\right)}^{2}}$$

After the Fisher score was calculated for each feature, the features were then ranked in descending order according to the Fisher score.

(2)MIFS

The MI [39] indicates the degree of correlation between two variables, i.e., the change degree of a variable when another variable is changed. There are three types of MI calculations in total according to the type of variable: between discrete and discrete data, between continuous and continuous data, and between discrete and continuous data. The MI between discrete and discrete data is calculated as shown in Equation (8). The MI between continuous and continuous data is calculated as shown in Equation (9).
(8)$$MI\left(X,Y\right)=\underset{y\in Y}{\sum }\underset{x\in X}{\sum }p\left(x,y\right)\log\left(\frac{p\left(x,y\right)}{p\left(x\right)p\left(x\right)}\right)$$
(9)$$MI\left(X,Y\right)=\int_{Y}^{}\int_{X}^{}p\left(x,y\right)\log\left(\frac{p\left(x,y\right)}{p\left(x\right)p\left(x\right)}\right)dxdy$$

In Equation (8) p(x,y) is the joint probability distribution function of X and Y, p(x) and p(y) are the marginal probability distribution functions of X and Y, respectively. In Equation (9) p(x,y) is the joint probability density function of X and Y, p(x) and p(y) are the marginal probability density functions of X and Y, respectively.

A novel calculation method of MI between discrete and continuous data was proposed by Ross [39] in 2014 and is widely used. He first finds the k-th closest neighbor to point i among those $N_{x_{i}}$ data points whose value of the discrete variable equals $x_{i}$ using some distance metric. Define d as the distance to this k-th neighbor. Then count the number of neighbors $m_{i}$ in the full dataset that lies within distance d to point i (including the k-th neighbor itself). Based on $N_{x_{i}}$ and $m_{i}$ calculate the MI by Equation (10).
(10)$$MI\left(X,Y\right)=\psi \left(N\right)-\langle \psi (N_{x_{i}})\rangle +\psi \left(k\right)-\langle \psi (m_{i})\rangle$$
where, ψ(·) is the digamma function, 〈⋯〉 means calculating the average value. In Equation (10) larger k-values lead to lower sampling error but higher coarse-graining error, we set the k as 3. In this paper, take each feature as a variable, and the label as another variable to calculate the correlation between each feature and the label. Features are sorted in descent according to the MI, the first S features are selected to form the new feature subset, and this feature selection method is called MIFS.

(3)VTFS

Divide the same features of all samples into a group and calculate the variance according to Equation (11) which is recorded as $s^{2}$.
(11)$$s^{2}=\frac{1}{n}\overset{n}{\underset{i=1}{\sum }}{\left(x_{i}-\bar{x}\right)}^{2}$$
where n represents the number of $x_{i}$, $\bar{x}$ represents the average value of $x_{i}$, a lower variance indicates a more consistent feature between samples. By default, a feature with zero variance will be eliminated, as there is no difference in feature between samples, which is an ineffective feature. Calculating the variance of the same feature for all samples has been the most widely utilized in feature selection. Features are sorted in descending order of the variance. Generally, it is necessary to define a threshold to eliminate the features which have a variance below the threshold. In this paper, to compare with other feature selection methods, the threshold is not defined, and the features are sorted by variance. Finally, the first S features are selected to form the new feature subset, and this feature selection method is called VTFS.

(4)F-testFS

F-test feature selection (F-testFS) [41] method ranks the features in descending order according to the f-score (Equation (12)) which calculates the ratio of the variance (Equation (11)) between categories and the variance within a category for a feature. A greater f-score means that the distance within the category is less and the distance between the category is more. The f-score is given by Equation (12).
(12)$$f-\mathrm{score}=\frac{S_{t}^{2}}{S_{c}^{2}}$$
where, $S_{c}^{2}$ represents the variance within a category, $S_{t}^{2}$ represents the variance between categories.

It is assumed that there are six samples, each of which contains one feature. Suppose that the six samples have three distribution states as shown in Figure 7, respectively. C_1_, C_2_ and C_3_ represent the three categories, respectively. In Figure 7, the variance of each category is calculated according to Equation (11) and then averaged, which is denoted as $S_{c}^{2}\_a$, $S_{c}^{2}\_b$ and $S_{c}^{2}\_c$, respectively. $S_{t}^{2}\_a$, $S_{t}^{2}\_b$ and $S_{t}^{2}\_c$ are calculated according to Equation (11), they represent the variance of the six samples in Figure 7 C_1_, C_2_ and C_3_, respectively.

(5)Chi2FS

The chi-square feature selection (chi2FS) [42] method ranks the features in ascending order according to the chi-square value (Equation (13)) The basic idea of the chi-square test lies in the degree of deviation between the actual observed value and the theoretical inferred value of statistical samples. The degree of deviation between the actual observed value and the theoretical inferred value determines the magnitude of the chi-square value. If the chi-square value is larger, the greater the degree of deviation between them is. If the two values are completely equal, the chi-square value is zero, indicating that the theoretical value is completely consistent with the actual value.
(13)$$\chi^{2}=\overset{k}{\underset{i=1}{\sum }}\frac{{\left(A_{i}-E_{i}\right)}^{2}}{E_{i}}$$
where $A_{i}$ represents the actual observed frequency of samples and $E_{i}$ represents the theoretical inferred frequency of samples. In this work, we utilized the scikit-learn of python to calculate the chi-square value between discrete and continuous data.

## 5. Results and Analysis

In this section, firstly, the number of iterations and the training rate of the dataset have been determined by utilizing the GNB classifier. Secondly, a k-NN classifier with different k values is used to select the number of features, and the features are sorted by utilizing the six feature selection methods. Finally, the selected features and 70% of the samples are utilized to select the optimal two hyperparameters of the SVM classifier, and the remaining 30% samples are utilized to test the fault diagnosis accuracy of the PMDCM model.

## 5.1. The Results Based on GNB Classifier

The fault diagnosis accuracy of the PMDCM’s model based on the GNB classifier is shown in Table 5. The number of iterations represents the number of times the model is trained by utilizing the GNB classifier, and the number of iterations ranges from 10 to 100 times, in the step of 10 times. The test rate ranges from 10% to 90% in the step of 10% number of features, the numbers of the feature are shown in Table 6; the test rate represents the proportion of the number of test samples in the total number of samples. As shown in Figure 8, for each number of test samples the models are carried out in ten different iterations. There are nine sets of the test sample in this work, a total of 10×9=90 models were established.

For the GNB classifier, there are no hyperparameters that need to be adjusted; the GNB classifier is suitable for determining the number of iterations and dividing the proportion of samples for the training dataset and testing dataset. The number of iterations and the dividing ratio of samples can be decided according to the former research in order to make the experiments more rigorous and make the number of iterations and the dividing ratio of samples more suitable for the PMDCMs’ dataset. The fault diagnosis models for PMDCM with the different number of iterations and the dividing ratio of samples are constructed by utilizing the GNB classifier. The experimental results are shown in Table 5.

To further analyze the diagnosis performance in Table 5, as can be seen in Figure 9, The smaller variance means the better stability of the model. Figure 9a shows the change in the variance value of diagnostic accuracy as the number of iterations increases. As can be seen from Figure 9a, when the number of iterations of the models is set to 10, the variance value of the diagnostic accuracy is 2.03, and when the number of iterations of the models is set to 20, the variance value of the diagnostic accuracy decreases from 2.03 to 0.09. When the variance value is 0.09, the model has sufficient stability. Therefore, the number of iterations is set to 20 in the next experiment. Figure 9b shows the change in the variance value of diagnostic accuracy as the test rate increases. When the proportion of samples utilized for the test increases from 10% to 20%, the variance value decreases from 1.57 to 0.24. When the proportion of samples utilized for the test increases from 20% to 30%, the variance value decreases to 0.03, which is close to 0. The diagnosis accuracy with small fluctuation space. Therefore, for the rest experiments, 70% of the samples are used for training the model, and 30% of the samples are used for testing the model.

## 5.2. The Results Based on k-NN Classifier

For machine learning models, decreasing the number of features by selecting features is the direct method to improve the training efficiency of models. The experimental process of feature selection by utilizing the six feature selection methods by utilizing the k-NN classifier with different k values can be seen in Figure 10. Firstly, the extracted features are normalized by utilizing the Min-Max normalization. Secondly, the six feature selection methods (in Section 4.6) are utilized to select features, and the feature number of each model increases from 10% number features to 100% number features in the 10% number features of the dataset. The number of extracted features from motor current signals is shown in Table 7, the number of features starts at 34 and ends at 340. Finally, the k-NN classifier is utilized to establish motor fault diagnosis models for the different number of features, the most important hyperparameter is the number of nearest neighbors for the k-NN classifier. In order to carry out a detailed experimental comparison, this paper set the k value of the k-NN classifier models in 3, 5, 7 and 9, respectively, and utilizes the six feature selection methods. Euclidean distance is utilized to calculate the distance between samples. For each feature selection method establish 10 models, the number of features of each model is listed in Table 7. The samples of each category are randomly divided into 70% and 30%, 1200×70%=840 is the number of samples used for the training model, and the remaining 1200×30%=360 number of samples are used for testing the performance of the models. To obtain a more stable performance, each model is iterated for 20 rounds, for each iteration, the samples will be redivided randomly, and the average performance is adopted as the performance of models.

Figure 11a–d shows the fault diagnosis results of setting the k value of the k-NN classifier to 3, 5, 7 and 9, respectively. By utilizing six feature selection methods. In Figure 11a, when the k value of the k-NN classifier is set to 3, the best fault diagnosis accuracy of 97.84% can be achieved when utilizing the VTFS method to select 170 features and the proposed GVFS method to select 238 features. In Figure 11b–d, the common result is that when utilizing the 238 features selected by utilizing the proposed GVFS method, the model can achieve the best fault diagnosis accuracy, 97.54%, 97.39% and 97%, respectively. When the 136 selected features are utilized to construct the fault diagnosis model, the VTFS method can achieve the best fault diagnosis accuracy. In order to further improve the fault diagnosis accuracy, the 136 (Utilizing the VTFS method) and 238 (Utilizing the GVFS method) selected features will be utilized to construct the fault diagnosis model.

As can be seen from Figure 11, when the k value of k-NN is set to 3, 5, 7 and 9, the fault diagnosis accuracy of PMDCMs have the same change trend. Therefore, we only analyze the results when the k value is set to 7 in detail. As shown in Figure 11c, as the number of features increases from 34 to 204 when FSFS and MIFS methods are utilized, the fault diagnosis accuracy of PMDCMs increases from 91.92% to 95.58% and from 92.82% to 96.49%, respectively, then with the increase of the feature number, the fault diagnosis accuracy of PMDCMs decreases in a fluctuating way. When the F-testFS method is utilized to select the feature, the fault diagnosis accuracy of PMDCMs increases from 88.76% to 96.43 gradually as the number of features increases from 34 to 272; when all features are utilized to establish the fault diagnosis model, the fault diagnosis accuracy decreases to 95.12%. When the number of features increases from 34 to 272 the fault diagnosis accuracy of PMDCMs increases from 90.1% to 96.21% in a fluctuating way by utilizing the Chi2FS method; when all features are utilized to establish the fault diagnosis model, the fault diagnosis accuracy decreases to 95.01%. As the number of features increases from 34 to 170, when the VTFS method is utilized, the fault diagnosis accuracy of PMDCMs increases from 90.11% to 97.01%; then with the increase of the feature number, the fault diagnosis accuracy of PMDCMs decreases in a fluctuating way. When utilizing the proposed GVFS method to construct the fault diagnosis models for PMDCMs, the number of features increase from 34 to 238; the fault diagnosis accuracy of PMDCMs increased from 87.11% to 97.39%, with the number of features increasing from 238 to 340, the fault diagnosis accuracy of PMDCMs decreased from 97.39% to 95.51%.

## 5.3. The Results Based on SVM Classifier

To further improve the fault diagnosis accuracy of PMDCMs, under the condition of determining the number of features. The SVM classifier is utilized to construct the PMDCM’s fault diagnosis model for 136 features selected by VTFS and 238 features selected by GVFS, respectively. As for the SVM-based fault diagnosis model of PMDCMs, there are two hyperparameters that affect the performance of the SVM classifier with the Gaussian kernel function, i.e., the penalty parameter C and the kernel parameter $\sigma^{2}$.

The process of PMDCM’s fault diagnosis by utilizing the SVM classifier is shown in Figure 12, the grid-search method is utilized to determine the two hyperparameters (C and $\sigma^{2}$) of the SVM. The grid-search method for the SVM optimization of the hyperparameters is carried out in a two-dimensional space: $\left(C_{i},\;\sigma_{i}^{2}\right)\in \left[2^{-8},\;2^{8}\right]\times \left[2^{-8},\;2^{8}\right]$. The grid size $\Delta C_{i}$ and $\Delta \sigma_{i}^{2}$ are set to $2^{1}$ and $2^{1}$, respectively. In the grid-search method, intersections in the grid space are formed by the combination of $C_{i}$ and $\sigma_{i}^{2}$, a total of 17×17=289 intersections. Each intersection corresponds to a fault diagnosis accuracy of PMDCMs. The samples of each category are randomly divided into 70% and 30%, 1200×70%=840 samples are used to search the optimal hyperparameters of the SVM classifier, and the remaining 1200×30%=360 samples are used to test the fault diagnosis accuracy of the PMDCMs. For each fault diagnosis, the accuracy of PMDCM’s model is checked by utilizing the combination of $C_{i}$ and $\sigma_{i}^{2}$; the 10-fold cross-validation [43] is utilized while the aim is to obtain stable fault diagnosis accuracy. The process of 10-fold cross-validation can be seen in Figure 13. The dataset is randomly divided into ten equal subsets; one of them is adopted as the ‘test set’, and the other nine subsets are taken as the ‘training set’. The ‘training set’ is utilized to construct the GVFS model, and the ‘test set’ is utilized to predict the evaluations. The model is carried out ten times, and the average prediction evaluation of the ten models is taken as the final evaluation of the model.

The fault diagnosis accuracies of PMDCMs are shown in Figure 14. Figure 14a shows fault diagnosis accuracies of PMDCMs by utilizing the different combinations of $C_{i}$ and $\sigma_{i}^{2}$, when utilizing the 136 features selected by utilizing the VTFS method. A total of 17×17=289 results are obtained. As shown in Figure 14a, the fault diagnosis accuracy of the area enclosed by the four points $(C_{i},\;\sigma_{i}^{2})=\left(2^{8},2^{1}\right),\left(2^{0},2^{1}\right),\left(2^{1},2^{0}\right),\left(2^{6},2^{-8}\right)$ is greater than 98%. We set the center point $\left(C=2^{5},\;\sigma^{2}=2^{-2}\right)$ of the enclosed area as the optimal hyperparameter of the SVM-based fault diagnosis model by utilizing the 136 features selected by utilizing the VTFS method. Figure 14b shows fault diagnosis accuracies of PMDCMs by utilizing the different combinations of $C_{i}$ and $\sigma_{i}^{2}$, when utilizing the 238 features selected by utilizing the GVFS method. A total of 17×17=289 results are obtained. As shown in Figure 14b, the fault diagnosis accuracy of the area enclosed by the four points $(C_{i},\;\sigma_{i}^{2})=\left(2^{8},2^{-1}\right),\left(2^{0},2^{-1}\right),\left(2^{2},2^{-2}\right),\left(2^{7},2^{0}\right)$ is greater than 98%. We set the center point $\left(C=2^{6},\;\sigma^{2}=2^{-3}\right)$ of the enclosed area as the optimal hyperparameter of the SVM-based fault diagnosis model by utilizing the 238 features selected by utilizing the GVFS method.

Figure 15a shows the fault diagnosis result of the 360 PMDCM samples with the 136 selected features by the VTFS method. First, the 1200 PMDCM samples are randomly divided into the proportion of 70% (840 samples) and 30% (360 samples), and then 70% of the divided samples are used for training the SVM-based fault diagnosis model with hyperparameters $\left(C=2^{5},\;\sigma^{2}=2^{-2}\right)$, and finally, use the remaining 30% samples to test the performance of the model. Figure 15b shows the fault diagnosis result of the 360 PMDCM samples with the 238 selected features by the GVFS method. First, the 1200 PMDCM samples are randomly divided into the proportion of 70% (840 samples) and 30% (360 samples), and then 70% of the divided samples are used for training the SVM-based fault diagnosis model with hyperparameters $\left(C=2^{6},\;\sigma^{2}=2^{-3}\right)$; finally, use the remaining 30% samples to test the performance of the model.

In Figure 15a,b, the horizontal axis represents the prediction state of PMDCMs, and the longitudinal axis represents the true state of PMDCMs. The state of PMDCMs faults is divided into six categories, each category has 60 test samples, and a total of 6×6=360 test samples. It can be seen from the two confusion matrices that most samples were correctly classified; however, in Figure 15a four samples were misclassified, and in Figure 15b two samples were misclassified, with the fault diagnosis accuracy of 98.89% and 99.44%, respectively.

As mentioned in the introduction, in order to improve the fault diagnosis efficiency of PMDCMs on the premise of ensuring the fault diagnosis accuracy of PMDCMs, this paper proposes a supervised filter feature selection method based on Gaussian probability density to reduce the number of features to improve the fault diagnosis efficiency of PMDCMs and compares the proposed feature selection method with the other five filter feature selection methods. GNB, k-NN and SVM classifiers are used to construct the fault diagnosis model of PMDCMs successively. First, the GNB classifier is used to determine the number of iterations of the experiment and the proportion of training samples. As shown in Figure 9 and Table 5, the variance of fault diagnosis accuracy is close to 0, when the number of iterations exceeds 20. Similarly, when the ratio of training samples to test samples is kept at 7 to 3, the fault diagnosis model of PMDCMs stays at a stable classification accuracy. Then the k-NN classifier is used to determine the optimal feature number. The experimental results are shown in Figure 11. When the k value of the k-NN classifier is set to 3, 5, 7 and 9, and the number of features selected by the proposed GVFS method is set to 238, the model can achieve the optimal fault diagnosis accuracy of 97.89%, 97.54%, 97.39% and 97%, respectively. Finally, of the 238 features selected by the proposed GVFS method, the SVM classifier is utilized to further improve the fault diagnosis accuracy of PMDCMs, so that the fault diagnosis accuracy is increased from 97.89% to 99.44%.

## 6. Conclusions and Future Work

In this work, a novel supervised feature selection method has been proposed, namely GVFS, to reduce the dimensional of the features based on GPDF for fault diagnosis of PMDCMs. The main conclusions are as follows:
(1)The proposed GVFS method has a better performance than the other five filter feature selection methods.(2)The iteration of the experiment can effectively reduce the fluctuation of PMDCM’s fault diagnosis accuracy.(3)The extracted features of PMDCM’s existence redundancy features and some features have a negative effect.

The effectiveness of the presented GVFS method is validated through the fault diagnosis experiment of PMDCMs. This paper provides theoretical guidance for feature selection and fault diagnosis of PMDCMs. In future work, the proposed GVFS in this paper will extend as a classifier and as a method to identify the feature weights.

## Figures and Tables

**Figure 1 sensors-22-07121-f001:**
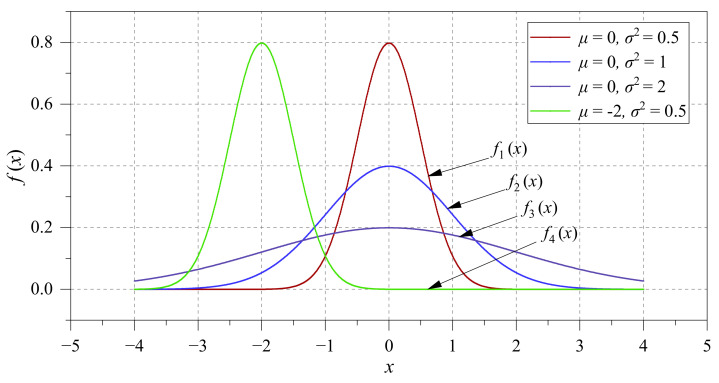
Four kinds of Gaussian probability density function.

**Figure 2 sensors-22-07121-f002:**
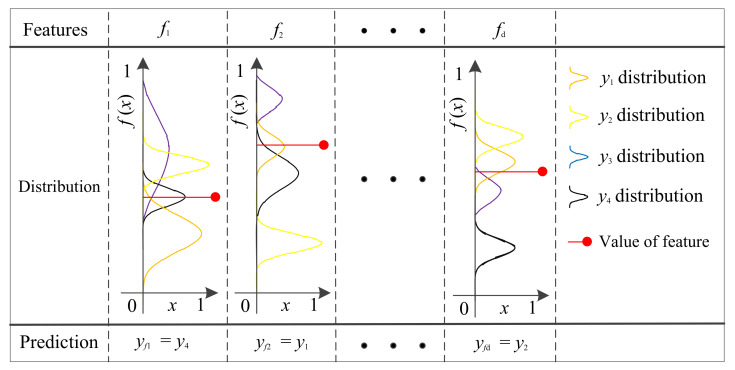
Diagram of feature classification based on Gaussian probability density function.

**Figure 3 sensors-22-07121-f003:**
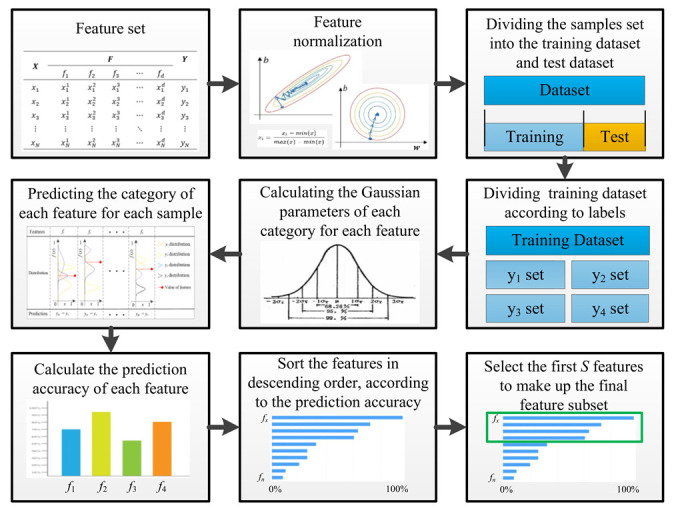
The process of the proposed GVFS method.

**Figure 4 sensors-22-07121-f004:**
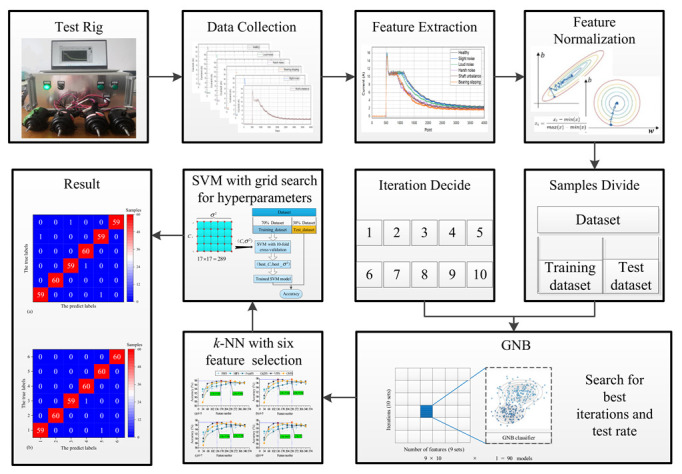
The framework of the experimental.

**Figure 5 sensors-22-07121-f005:**
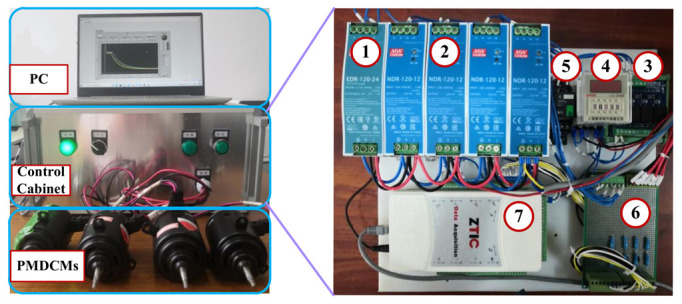
The fault diagnosis platform for PMDCMs. (1) 24 V power supply, (2) 12 V power supply, (3) Front-end relay, (4) Timer relay, (5) 4-channel relay, (6) Sampling resistance, (7) Data acquisition card.

**Figure 6 sensors-22-07121-f006:**
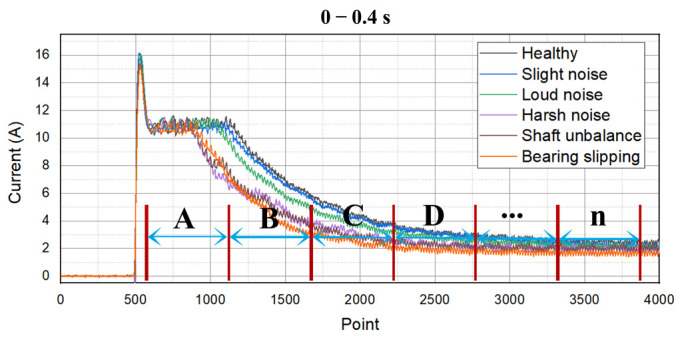
Multi-segment feature extraction. (A, B, C, D, ⋯, n represent the multi-segment signal.)

**Figure 7 sensors-22-07121-f007:**
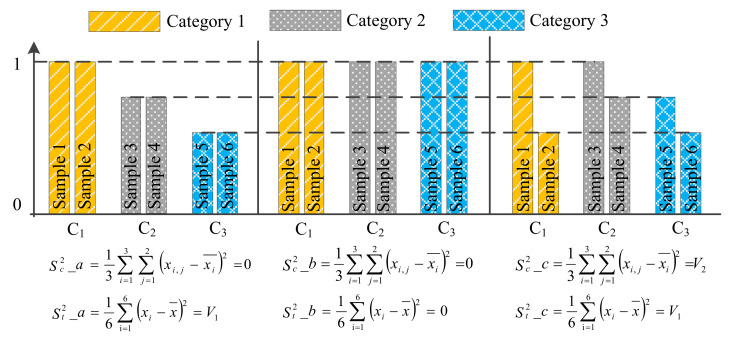
Three distribution states of the same feature for six samples.

**Figure 8 sensors-22-07121-f008:**
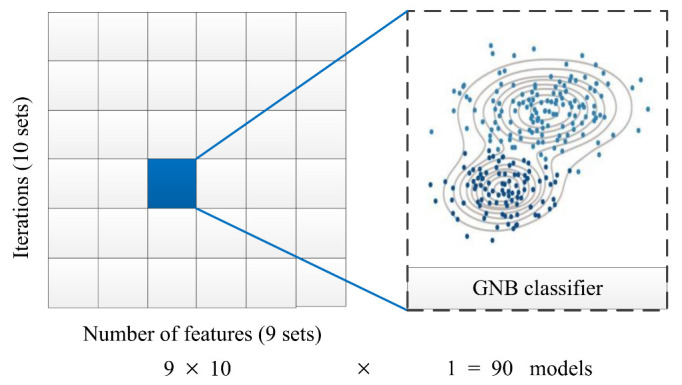
The process of selecting the number of features and iterations.

**Figure 9 sensors-22-07121-f009:**
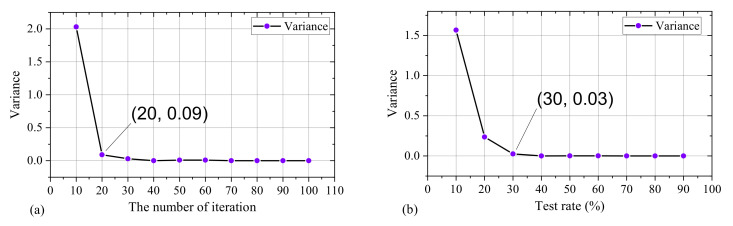
The variance value of diagnosis accuracy for different iterations and test rate, (**a**) The variance value of diagnosis accuracy for different iterations, (**b**) The variance value of diagnosis accuracy for different test rate.

**Figure 10 sensors-22-07121-f010:**
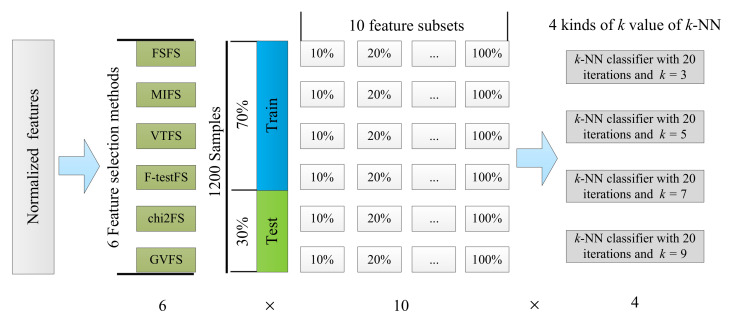
The experimental process of feature selection by utilizing the six feature selection methods and by utilizing the k-NN classifier with different k values.

**Figure 11 sensors-22-07121-f011:**
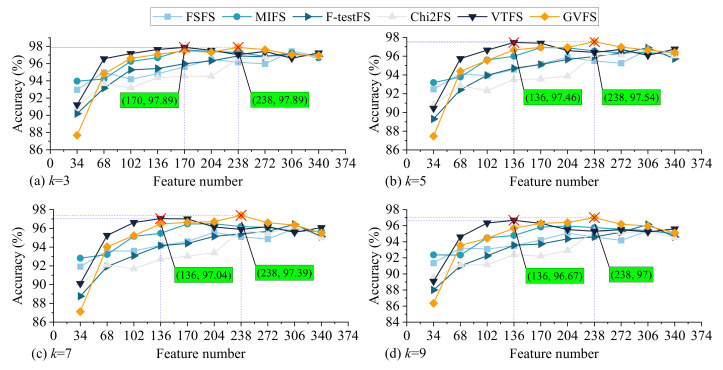
The fault diagnosis accuracy by utilizing the k-NN classifier with different k value.

**Figure 12 sensors-22-07121-f012:**
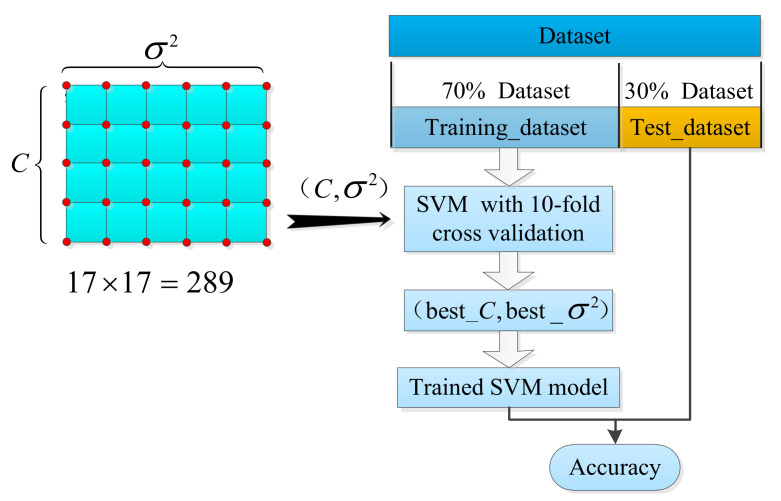
The process of PMDCM’s fault diagnosis by utilizing the SVM classifier.

**Figure 13 sensors-22-07121-f013:**
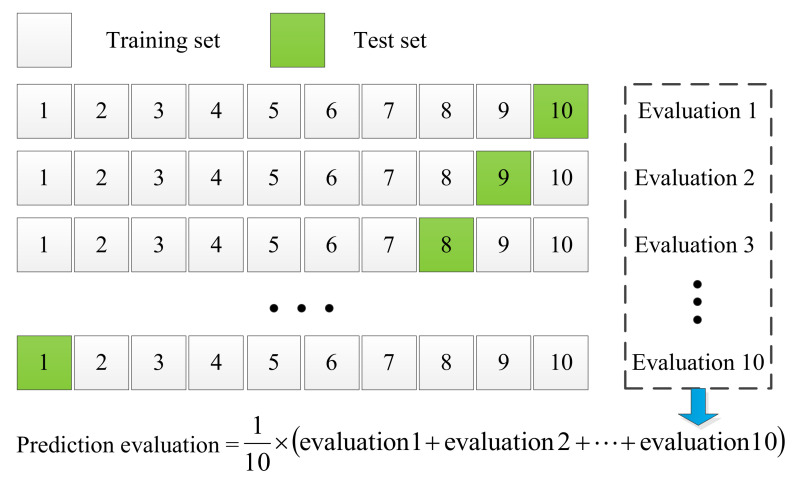
The process of 10-fold cross-validation.

**Figure 14 sensors-22-07121-f014:**
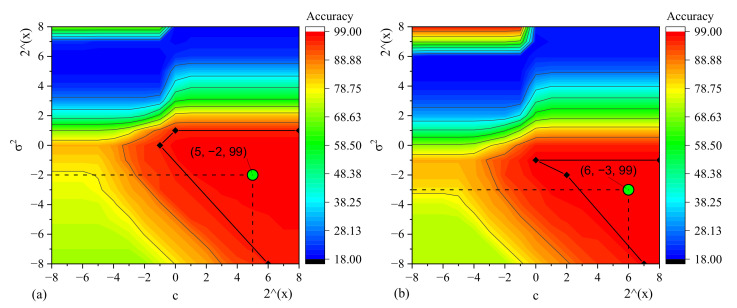
The fault diagnosis accuracies of PMDCMs by utilizing the SVM classifier with different hyperparameters, (**a**) with the 136 selected features by the VTFS method, (**b**) with the 238 selected features by the GVFS method.

**Figure 15 sensors-22-07121-f015:**
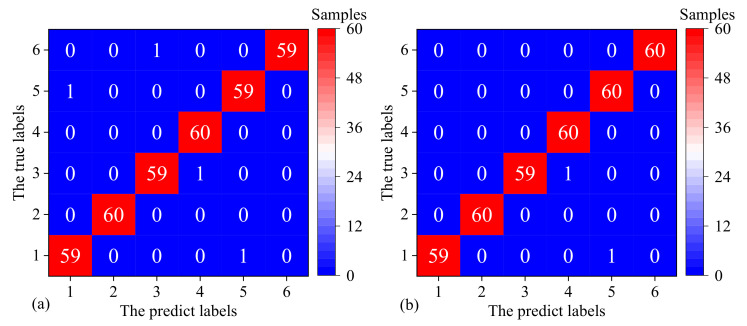
The fault diagnosis confusion matrixes by utilizing the SVM classifier of the 360 PMDCM samples (each category has 60 samples) with the 136 selected features by the VTFS method (**a**), and with the 238 selected features by the GVFS method (**b**).

**Table 1 sensors-22-07121-t001:** Feature structure of a dataset.

X	F					Y
	$f_{1}$	$f_{2}$	$f_{3}$	⋯	$f_{d}$	
$x_{1}$	$x_{1}^{1}$	$x_{1}^{2}$	$x_{1}^{3}$	⋯	$x_{1}^{d}$	$y_{1}$
$x_{2}$	$x_{2}^{1}$	$x_{2}^{2}$	$x_{2}^{3}$	⋯	$x_{2}^{d}$	$y_{2}$
$x_{3}$	$x_{3}^{1}$	$x_{3}^{2}$	$x_{3}^{3}$	⋯	$x_{3}^{d}$	$y_{3}$
⋮	⋮	⋮	⋮	⋱	⋮	⋮
$x_{N}$	$x_{N}^{1}$	$x_{N}^{2}$	$x_{N}^{3}$	⋯	$x_{N}^{d}$	$y_{N}$

**Table 2 sensors-22-07121-t002:** Type and function description of the control cabinet parts.

No.	Name	Type	Function Description
1	24 V power supply	Mean Well EDR-120-24	220 V AC to 24 V DC
2	12 V power supply	Mean Well NDR-120-12	220 V AC to 12 V DC
3	Front-end relay	BMZ 02R1-E	Switch of the circuit
4	Timer relay	ANLY AH3-2	Trigger delay: 0~10s
5	4-channel relay	Schneider Electric RX4AB2BD	Switch of the motors
6	Sampling resistance	TO247-100W-0.2	High-frequency-based digital sampling
7	Data acquisition card	ZTIC EM9118B	Collection of voltage signals

Note: AC represents alternating current, DC represents direct current.

**Table 3 sensors-22-07121-t003:** Categorization for healthy and faulty PMDCMs.

Class	Actual Status	Number of PMDCMs
1	Healthy	3
2	Slight noise	1
3	Loud noise	2
4	Harsh noise	1
5	Shaft unbalance	1
6	Bearing slipping	1

Note: PMDCM represents permanent magnet DC motor.

**Table 4 sensors-22-07121-t004:** The extracted time domain features.

Index	Features	Expression
1	Mean (μ)	$\mu =E\left(x_{i}\right)$
2	Root Mean Square (RMS)	$x_{RMS}={\left\{E\left(x_{i}^{2}\right)\right\}}^{1/2}$
3	Maximum (Max)	$x_{Max}=max\left(x_{i}\right)$
4	Peak to Valley (PV)	$x_{PV}=\max\left(x_{i}\right)-min\left(x_{i}\right)$
5	Standard deviation (Std)	$x_{Std}=\sigma ={\left\{E\left[{\left(x_{i}-\mu \right)}^{2}\right]\right\}}^{1/2}$
6	Skewness (Ske)	$x_{Ske}=E\left\{{\left[\left(x_{i}-\mu \right)/\sigma \right]}^{3}\right\}$
7	Kurtosis (Kur)	$x_{Kur}=E\left\{{\left[\left(x_{i}-\mu \right)/\sigma \right]}^{4}\right\}$
8	Shape Factor (SF)	$x_{SF}=x_{RMS}/\mu$
9	Crest Factor (CF)	$x_{CF}=x_{Max}/x_{RMS}$

**Table 5 sensors-22-07121-t005:** The diagnosis accuracy of the PMDCM based on GNB classifier.

		The Number of Iterations										
		10	20	30	40	50	60	70	80	90	100	Var
Test rate	10%	90.83	87.42	87.08	86.95	86.88	86.84	86.81	86.79	86.78	86.76	1.57
	20%	88.33	87.18	86.97	86.88	86.83	86.8	86.78	86.76	86.75	86.74	0.24
	30%	87.22	86.89	86.8	86.76	86.74	86.73	86.72	86.71	86.71	86.7	0.03
	40%	86.77	86.71	86.7	86.69	86.68	86.68	86.68	86.68	86.67	86.67	0.00
	50%	86.52	86.59	86.61	86.62	86.63	86.64	86.64	86.64	86.65	86.65	0.00
	60%	86.54	86.59	86.61	86.63	86.63	86.64	86.64	86.64	86.65	86.65	0.00
	70%	86.61	86.63	86.64	86.65	86.65	86.65	86.65	86.65	86.65	86.66	0.00
	80%	86.71	86.69	86.68	86.68	86.68	86.67	86.67	86.67	86.67	86.67	0.00
	90%	86.66	86.66	86.66	86.66	86.66	86.66	86.66	86.66	86.66	86.66	0.00
	Var	2.03	0.09	0.03	0.01	0.01	0.01	0.00	0.00	0.00	0.00	

Note: Var represents the variance value.

**Table 6 sensors-22-07121-t006:** The number of features.

Test Rate	10%	20%	30%	40%	50%	60%	70%	80%	90%
Test sample number	120	240	360	480	600	720	840	960	1080

**Table 7 sensors-22-07121-t007:** The number of features.

Feature Rate	10%	20%	30%	40%	50%	60%	70%	80%	90%	100%
Feature numbers	34	68	102	136	170	204	238	272	306	340

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
