# Peer review of "A Novel Supervised Filter Feature Selection Method Based on Gaussian Probability Density for Fault Diagnosis of Permanent Magnet DC Motors"

_sensors, 2022, doi:10.3390/s22197121_

Round 1

Reviewer 1 Report

(1) The detailed technical proceudres of th proposed method should be clearly presented in Fig. 3.

(2)Why the iterating process is needed  should be clearly shown and explained.

(3) The proposed  method only serves as an important step of the the fault diagnosis method. The performance of the the proposed method should be evaluated independent of the classification methods, like SVM,KNN etc. 

Reviewer 2 Report

This paper presents a novel supervised filter feature selection method for reducing data dimension by employing the Gaussian probability density function (GPDF), named Gaussian vote feature selection (GVFS). In order to further improve the quality of the paper,  the following major comments are required to be considered.   

1. The Abstract is too long to catch the key contents of this manuscript. The authors are required to rewrtite it as brief as possible.

2. The motivations and the main contributions of this work should be discussed and highlighted more clearly. 

3. Since only the Gaussian distribution is studied in this article, the discussions about other probability distributions like Gamma distribution seems insufficient. Recently, some related results studying the Gamma distribution are given as:  Probability-density-dependent load frequency control of power systems with random delays and cyber-attacks via circuital implementation;  Synchronization of delayed fuzzy neural networks with probabilistic communication delay and its application to image encryption;  H∞ weighted integral event-triggered synchronization of neural networks with mixed delays; Event-triggered  control of networked control systems with distributed transmission delay.  

4. More comparison results with some existing methods should be provided to show the advantages of the proposed method. 

5. To make the paper more clear for readers, the detailed information of the two figures in Fig. 13 are required to added in figure title.

6. There are some grammar errors and typos, please check the presentation carefully. 

Reviewer 3 Report

This paper present a new filter feature selection method for fault diagnosis of permanent magnet DC motors. According to the authors, the proposed solution improves diagnostic efficiency and decreases computing costs.

The paper has good archival value and is valuable to practicing engineers. 

I present below some comments and observations:

In my opinion, and as the topic addressed in the paper is related to fault diagnosis, the authors should present what are the main types of faults that can occur in PMDCMs, their distribution, the fault mechanism and how they manifest themselves.

Authors present a very complete state of the art on feature selection methods, however, the paper lacks a literature review on fault diagnosis in permanent magnet DC motors.

Authors should describe the experience in more detail, namely how the different fault conditions were implemented.

The proposed solution (GVFS method) is compared with the other five feature selection methods, however, authors should also compare the proposed solution with existing fault diagnosis in permanent magnet DC motors.

In the 2nd line of section 5.1, I believe there is an error regarding the identification of the table (table 6 and not 3).

In the 7th line of section 5.2, I believe there is an error regarding the identification of the table (table 7 and not 3).

Authors should also discuss the applicability of the proposed solution in a commercial application.

Round 2

Reviewer 2 Report

No further comments,it can be published now.

Reviewer 3 Report

The authors answered all questions clearly and introduced the proposed suggestions.

In my opinion, the paper has been improved and merits publication.